# Role of Hsp70 in Post-Translational Protein Targeting: Tail-Anchored Membrane Proteins and Beyond

**DOI:** 10.3390/ijms24021170

**Published:** 2023-01-06

**Authors:** Shu-ou Shan

**Affiliations:** Division of Chemistry and Chemical Engineering, California Institute of Technology, Pasadena, CA 91125, USA; sshan@caltech.edu

**Keywords:** Hsp70, J-domain protein, molecular chaperone, membrane protein, protein targeting, GET pathway, endoplasmic reticulum, mitochondria

## Abstract

The Hsp70 family of molecular chaperones acts as a central ‘hub’ in the cell that interacts with numerous newly synthesized proteins to assist in their biogenesis. Apart from its central and well-established role in facilitating protein folding, Hsp70s also act as key decision points in the cellular chaperone network that direct client proteins to distinct biogenesis and quality control pathways. In this paper, we review accumulating data that illustrate a new branch in the Hsp70 network: the post-translational targeting of nascent membrane and organellar proteins to diverse cellular organelles. Work in multiple pathways suggests that Hsp70, via its ability to interact with components of protein targeting and translocation machineries, can initiate elaborate substrate relays in a sophisticated cascade of chaperones, cochaperones, and receptor proteins, and thus provide a mechanism to safeguard and deliver nascent membrane proteins to the correct cellular membrane. We discuss the mechanistic principles gleaned from better-studied Hsp70-dependent targeting pathways and outline the observations and outstanding questions in less well-studied systems.

## 1. Hsp70: An Allosteric Machine with Diverse Functions in Protein Biogenesis

Proper proteostasis is essential to all cells and relies critically on a network of molecular chaperones that ensure the correct folding, localization, and quality control of all cellular proteins [1,2]. Among the chaperone network, the universally conserved 70-kDa heat shock proteins (Hsp70s) comprise a central hub that plays a dominant role in regulating proteome homeostasis [1,2,3,4]. Cytosolic Hsp70s are abundant and associate with numerous newly synthesized proteins, during and after their synthesis by the ribosome, to facilitate their folding. Hsp70s recognize client proteins at multiple stages of their life cycle, from the unfolded state, partial folding intermediates, to protein assemblies and aggregates [5]. Hsp70 homologs that reside in cellular organelles, such as the Binding immunoglobulin protein (BiP) in the endoplasmic reticulum (ER) lumen and mitochondrial Hsp70 in the mitochondrial matrix, facilitate protein translocation across organellar membranes by binding precursor proteins at the luminal side of translocases [6,7]. They also serve as decision points at which client proteins are triaged to additional protein biogenesis and quality control machineries [3,8]. As such, Hsp70s impact diverse aspects of protein homeostasis, from de novo folding, disaggregation, remodeling of protein complexes, to trafficking and degradation.

The function of Hsp70 relies on the ability of its substrate binding domain (SBD) to bind short extended segments of sequences enriched in hydrophobic residues (often flanked by basic amino acids) that are exposed on client proteins [5,8,9,10]. Client affinity is regulated by the ATPase cycle in the nucleotide binding domain (NBD) of Hsp70 and is tightly coupled to interaction with co-chaperones [1,2,3,8]. In the ATP-bound state, the α-helical lid of the SBD (SBDα) packs against the NBD and is away from the peptide binding site in the SBD (SBDβ; Figure 1A), allowing rapid client binding and dissociation. Client proteins and a class of cochaperones, termed Hsp40 or J-domain proteins (JDPs), stimulate ATP hydrolysis on Hsp70 [11,12]. This converts Hsp70s to the ADP-state that binds client proteins with higher affinity and kinetic stability, in part due to closing of the α-helical lid over SBDβ (Figure 1A). In this state, Hsp70 induces the local unfolding of client protein by clamping on the bound polypeptide and via entropic pulling mechanisms. ATP hydrolysis also drives the dissociation of the J-domain of JDP from ADP-bound Hsp70, for which it has low affinity. Another class of cochaperones, the nucleotide exchange factors (NEFs), facilitate client dissociation from Hsp70 by accelerating ADP release and, in some cases, directly contacting the Hsp70-SBD to drive client displacement (Figure 1A) [13,14,15,16]. These allosteric regulations allow Hsp70 to act as molecular machines that capture and protect unfolded segments on client proteins and, after some dwell time set by the ATPase cycle, to release client proteins in conformations that are more conducive to folding (Figure 1B, steps 2–3) [17,18]. In addition, JDPs harbor interaction motifs for client proteins or specific protein complexes [19,20]. Given that there are 22 different JDPs in yeast and 50 different JDPs in human cells [20], the JDP-Hsp70 pairing enables diverse client proteins in different conformational states to be remodeled by the Hsp70 machinery.

Moreover, Hsp70s evolved multiple protein interaction motifs that allow them to recruit and hand off client proteins to other chaperones and cochaperones, providing a versatile triaging mechanism via which client proteins are directed to their dedicated biogenesis or quality control pathways. For example, client proteins resistant to spontaneous or Hsp70-assisted folding are captured and folded by chaperonins, such as GroEL/ES in bacteria and mitochondria and TRiC in the eukaryotic cytosol (Figure 1B, path 4) [21,22,23]. Hsp70s, via a direct interaction with the M-domain of the Hsp100 family of chaperones, recruit these disaggregate machineries to protein aggregates and allosterically activate them for aggregate remodeling (Figure 1C, path 5), providing a mechanism to reactivate misfolded and aggregated proteins [24,25,26,27,28]. Eukaryotic Hsp70s further evolved a C-terminal extension that harbors a conserved motif with the amino acid sequence EEVD, which is recognized by a subclass of tetratricopeptide repeat (TPR) domains frequently found in heat shock cognate protein (HSC) cochaperones, including Hsp organizing protein (HOP in human and Sti1 in yeast) and C-terminus of HSC interacting protein (CHIP) [29,30,31]. The best studied example is Sti1/HOP, which organizes Hsp70 and Hsp90 into a multi-chaperone complex in which the folding of members of the kinase superfamily and hormone receptor are completed (Figure 1C, path 6) [32,33,34,35]. On the other hand, interaction of Hsp70 with the CHIP E3 ligase leads to ubiquitylation of the client protein and its degradation by the proteasome (Figure 1C, path 7). The functional diversity of Hsp70 is thus further expanded by its interaction with downstream interaction partners.

Notably, homologous TPR domains that interact with Hsp70 have been found in cochaperones and receptors involved in the targeting of proteins to multiple organelles, enabling Hsp70 to further participate in protein targeting pathways (Figure 1C, path 8). Below, we review recent progress in delineating the roles of Hsp70 in assisting the targeted delivery of membrane and organellar proteins and the lessons from these studies on the principles of chaperone-assisted, post-translational membrane protein targeting.

## 2. Overview of Protein Targeting and the Role of Hsp70

Integral membrane proteins (MPs) comprise over 30% of the proteins encoded by the genome and mediate numerous essential cellular processes including molecular transport, energy generation, signaling, and cell-to-cell communication. Over 1000 proteins in eukaryotic cells also reside in the lumen of the endoplasmic reticulum (ER), Golgi, lysosomes, peroxisomes, as well as the intermembrane space and matrix of mitochondria [36,37]. Compared to soluble proteins, the biogenesis of MPs and organellar proteins poses acute challenges to protein homeostasis in the cell. Before arrival at their appropriate cellular destination, newly synthesized MPs and organellar proteins must traverse the cytosol and, in some cases, multiple other non-native environments where they are prone to misfolding, aggregation, or degradation. This problem is particularly acute for MPs, on which the improper exposure of transmembrane domains (TMDs) will lead to rapid and irreversible aggregation [38,39]. In addition, the degeneracy of TMD–lipid interaction poses challenges to the fidelity of their insertion at the appropriate biological membrane, especially in eukaryotic cells that contain multiple membrane-enclosed organelles.

For these reasons, the majority of proteins destined to the endomembrane system in eukaryotic cells are initially targeted to the ER membrane in a co-translational mechanism [40,41,42,43]. These proteins generally contain an N-terminal ER-targeting signal, either as the first TMD of an MP or as a cleavable hydrophobic signal sequence on a secretory protein. The ER-targeting signals are recognized by the signal recognition particle (SRP) as soon as they emerge from the nascent polypeptide exit tunnel of the ribosome (Figure 2, path 1). Via interaction with the SRP receptor, the nascent proteins are delivered to the Sec61 translocation machinery and other insertases at the ER membrane for cotranslational translocation or insertion. By coupling protein synthesis to their targeting and insertion, the SRP pathway minimizes the misfolding and aggregation during the biogenesis of ER-bound membrane and secretory proteins. The readers are referred to [43,44,45,46] for comprehensive reviews on the SRP pathway.

A notable exception to this paradigm is tail-anchored membrane proteins (TAs), defined by a single TMD near the C-terminus. These proteins comprise 3–5% of the eukaryotic membrane proteome and mediate diverse cellular processes including protein translocation, vesicle fusion, apoptosis, and protein quality control [47,48,49]. As their C-terminal TMD is buried inside the ribosome during translation, it was predicted early on that TAs must use post-translational mechanisms for membrane targeting [50]. Multiple pathways for delivering TAs to the ER (Figure 2, path 2a) have been reported, including the guided entry of tail-anchored protein (GET) pathway [47,48,49] and the SRP-Independent (SND) pathway in yeast and mammalian cells [51]. In addition, the ER-membrane complex (EMC) can insert a subset of TAs into the ER [52]. However, a substantial number of TAs are destined for mitochondria (Figure 2, path 2b). In addition, the majority of mitochondrial proteins harbor distinct biophysical features and are targeted via post-translational mechanisms (Figure 2, path 3). While a dedicated targeting factor has not been identified for these proteins, an abundance of data indicates an integral role of Hsp70 in their proper biogenesis.

The participation of Hsp70 in the biogenesis of membrane and organellar proteins was initially suggested by the observation that depletion of the major cytosolic Hsp70 in yeast, Ssa1, severely compromised the secretion of prepro-α-factor and the import of a mitochondrial protein, the β-subunit of F_1_ ATPase (F1β) [53,54]. Similar observations were made with depletion or inactivation of the major cytosolic JDP in yeast, Ydj1 [53]. However, significant caution needs to be exercised in interpreting the results of Hsp70/JDP knockout, depletion, or inactivation experiments in cells. In addition to the functional overlap among multiple Hsp70 paralogs in a cell, Hsp70s are extensively involved in the folding and protection of the proteome; thus, the observed defects could arise from indirect effects, such as impaired folding or stability of key targeting and/or translocation machineries. Indeed, significant defects in the activity of established Hsp70 substrates could be observed as soon as 60 min after Ssa1 inactivation [55,56]. To minimize these potential complications, many subsequent studies of Hsp70 function in yeast used a strain that only expresses a temperature sensitive (ts) mutant of Ssa1, which can be inactivated within 5 min of shift to nonpermissive temperature [57] and thus allows the role of Hsp70 in nascent protein biogenesis to be examined prior to global perturbation of the proteome. Re-examination of protein import using this strain showed that the translocation of two out of six ER-destined proteins and one out of three mitochondrial proteins (including prepro-α-factor and F1β) were inhibited upon acute inactivation of Ssa1 [57]. These studies provide evidence for an important role of Ssa1 in the biogenesis of a subset of organellar proteins. Given the homology of Hsp70 paralogs, it is likely that the other cytosolic Hsp70s in yeast, Ssa2–Ssa4, play overlapping roles to that of Ssa1 in supporting protein translocation.

In the simplest model, cytosolic Hsp70s could enhance the biogenesis of membrane, organellar, and secretory proteins by protecting them from off-pathway interactions, such as misfolding and aggregation, and maintain them in a loosely folded conformation required for translocation across the membrane. However, accumulating data support a more direct involvement of Hsp70 in the targeted delivery of MPs and organellar proteins, based on the findings that it can associate with components of protein targeting or translocation pathways. An early example is the Sec62/63/71/72 tetrameric complex (Figure 3), which mediates SRP-independent, post-translational translocation of secretory proteins and GPI-anchored membrane proteins across the ER membrane [58,59,60]. Following early studies in yeast showing a requirement of cytosolic Hsp70s for preprotein translocation [54,57,58,61], crosslinking experiments showed that post-translational substrates associate with Hsp70 and other chaperones as soon as they are released from the ribosome and subsequently contact the tetrameric complex [62]. Importantly, the cytosolic domain of Sec72 contains a TPR domain that was found to bind the C-terminal EEVD motif of Ssa1 with weak affinity [63]. A crystal structure of this complex fused to the C-terminal peptide of Ssa1 revealed an interaction mode of the EEVD motif with the first three TPR motifs of Sec72, analogous to the interactions observed with HOP, Sgt2, and Tom71 [63]. Unexpectedly, the ribosome-associated Hsp70 in yeast, Ssb1, also bound to Sec72 via its NBD, and this interaction led to the inhibition of nucleotide exchange on Ssb1 [63]. Whether either of these interactions provides a path for delivering proteins to the ER membrane remains to be determined.

More recently, the mechanism by which Hsp70 assists in the ER targeting of TAs by the GET pathway was extensively studied; this pathway provided the best-established example of a direct role of Hsp70 in post-translational MP targeting and will be discussed first below. A large body of work also support a role of Hsp70 in delivering proteins to receptor sites on the mitochondria surface; this and the potential involvement of Hsp70 in other protein targeting pathways will be discussed next.

## 3. An Hsp70-Cochaperone Cascade Guides TAs to the ER

The GET pathway is responsible for delivering TAs with highly hydrophobic TMDs to the ER. Components of this pathway were initially identified in genetic interaction analyses in yeast, which uncovered strong epistatic interactions between five genes named Get1–Get5 [65,66]. Mammalian homologs or functional orthologs of all the GET proteins have subsequently been identified [47,48,49]. For clarity, we will refer to yeast proteins in sentence case, and mammalian proteins in uppercase.

Biochemical studies showed that Get3 (or its mammalian homolog TRC40) is a cytosolic ATPase that can bind the C-terminal TMD of model TAs, whereas Get1 and Get2 together form a receptor complex at the ER membrane for TA-loaded Get3 and also act as an insertase to mediate the membrane insertion of GET-dependent TAs (Figure 4A, step 6) [67,68]. The structure, dynamics, and activity of Get3 and Get1/2 have been extensively studied, providing a molecular model for how TAs are captured, protected, and delivered by Get3 to its ER receptors for insertion. The readers are referred to [47,48,49] for comprehensive reviews on the mechanism of these protein targeting and insertion machineries in the GET pathway.

Nevertheless, it soon became clear that TA capture by Get3 (or TRC40) is a facilitated process in the crowded cytosolic environment. Nascent TA released from the ribosome was poorly captured by partially purified TRC40 [69]. Biochemical reconstitutions by Wang et al. showed that Get4 and Get5, two additional proteins epistatically linked to Get3, form a scaffold complex that bridges between Get3 and an upstream cochaperone, Sgt2, to facilitate the transfer of TAs from Sgt2 to Get3 (Figure 4, step 5) [70]. Homologs of Sgt2, Get4 and Get5 were found in mammalian cells (SGTA, TRC35 and UBL4A, respectively) and were necessary and sufficient to reconstitute TA transfer from SGTA to TRC40 [71,72], indicating that the substrate loading mechanism via the Sgt2-to-Get3 transfer is conserved across eukaryotic cells.

**Figure 4 ijms-24-01170-f004:**
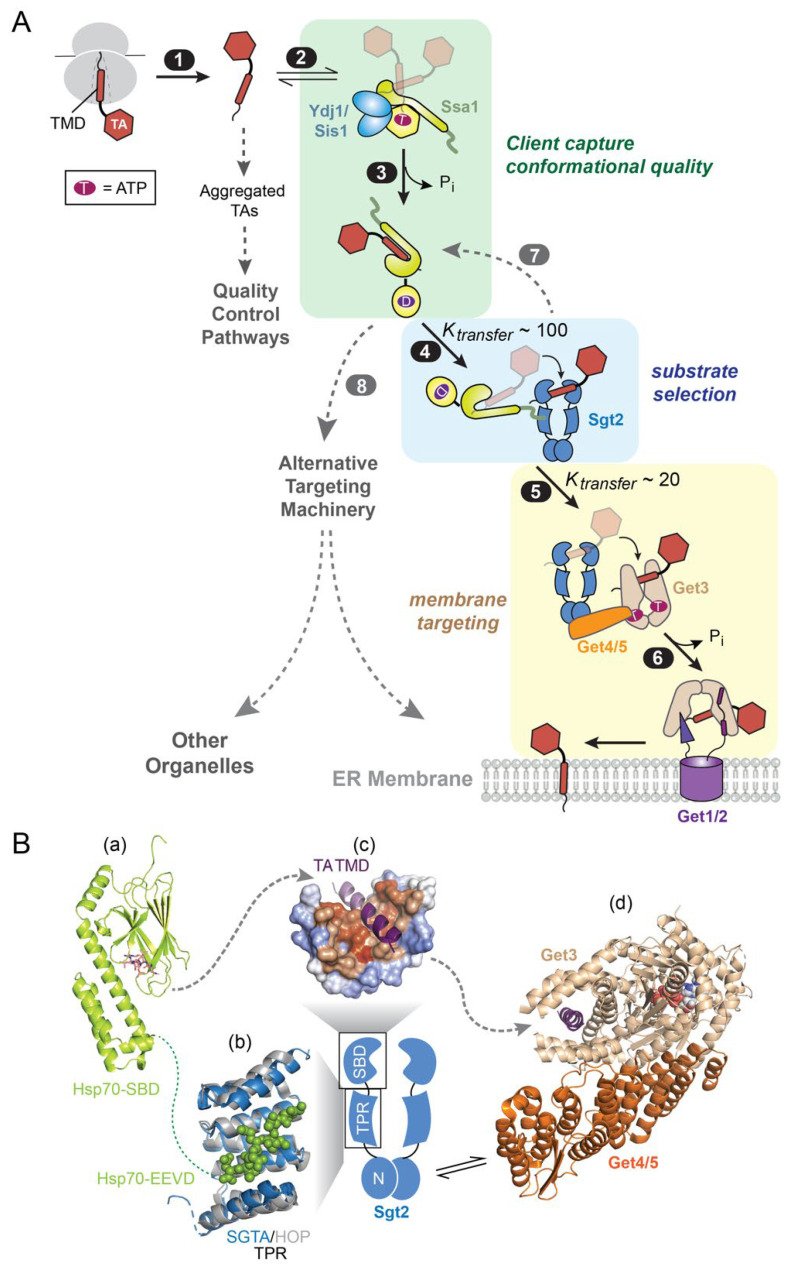
An Hsp70-cochaperone cascade mediates TA targeting to the ER via the GET pathway. (**A**) Scheme of the molecular steps in the GET pathway. TAs released from the ribosome (step 1) associate with the major cytosolic Hsp70 in yeast, Ssa1 (step 2). Two JDPs, Ydj1 and Sis1, catalyze ATP hydrolysis on Ssa1 and assist in substrate trapping (step 3). TA-bound Ssa1 associates with the cochaperone Sgt2 via its C-terminal EEVD motif, forming the first transfer complex in which TA is loaded onto Sgt2 (step 4). Get4/5 bridges between Sgt2 and the targeting factor Get3, forming the second transfer complex in which TA is relayed onto Get3 (step 5). TA-activated ATP hydrolysis on Get3 drives its dissociation from Get4/5 and association with the Get1/2 receptor complex at the ER membrane (step 6), via which the TA substrate is inserted. The dashed lines indicate that suboptimal TA substrates can be rejected by Sgt2 (step 7) and re-routed to alternative targeting pathways (step 8). (**B**) Summary of available structural information on the TA transfer complexes in the GET pathway. Panel (a), structure of the DnaK SBD bound to the NRLLLTG peptide (shown as *stick*; PDB 1dkz). Panel (b), the structure of the TPR domain of SGTA (*blue*; PDB 5lyn) is overlayed onto that of the TPR1 domain of HOP (*grey*) bound to the HSC70 C-terminal peptide GPTIEEVD (shown in *spacefill*; PDB 1elw). Panel (c), molecular model of the Sgt2/SGTA SBD bound to a TA-TMD (*purple*), shown in surface representation with hydrophobic residues in *orange* and hydrophilic residues in *blue.* Adapted from [73]. Panel (d), the structure of Get4/5 (*orange)* bound to Get3 (*tan*; PDB 4pwx) is overlayed onto that of Get3 bound to the TMD of the model TA Pep12 (*purple*; PDB 4xtr). The dotted line indicates unresolved flexible regions, and dashed arrows depict paths of TA transfer.

However, the way in which newly synthesized TAs are captured by Sgt2 remained a long-standing puzzle [74]. Sgt2 and SGTA by themselves are ineffective in capturing TAs in the soluble form, and attempts to directly load TA onto them led to extensively aggregated complexes [38]. For many years since the discovery of the GET pathway, generation of soluble, functional Sgt2•TA or SGTA•TA complexes relied on cell lysates that contain endogenous chaperone [71,75] or super-physiological Sgt2/SGTA concentrations [72]. The resolution to this dilemma was provided by biochemical studies showing that Hsp70 is responsible for the efficient and productive loading of TAs on Sgt2 (Figure 4A, steps 1–4) [38,76]. The major yeast cytosolic Hsp70, Ssa1, is much more effective in capturing TAs and protecting them from aggregation compared to Sgt2 (Figure 4A, steps 1–2) [38]. TA capture by Ssa1 could be further enhanced by Ydj1, the major cytosolic JDP in yeast, resulting in effective TA trapping on Ssa1 (Figure 4A, step 3) [76]. Moreover, Sgt2 (and SGTA) is characterized by a subclass of TPR domains frequently found in HSC cochaperones, including Sti1/HOP and CHIP. Five conserved residues in this TPR domain form a dicarboxylate clamp which, in the structure of HOP, recognizes a C-terminal EEVD motif in Hsp70, Hsp90, and Hsp100 (Figure 4B, structural panel b) [77,78]. Efficient transfer of TA substrates from Ssa1 to Sgt2 was observed in fluorescence and site-specific crosslinking assays and was dependent on the interaction of Hsp70 with the Sgt2 TPR domain (Figure 4A, step 4) [38,72,75,76]. Finally, Hsp70, Sgt2, Get4/5 and Get3 form the minimal components necessary and sufficient to reconstitute a soluble, translocation-competent Get3•TA targeting complex [38]. In support of an essential role of Hsp70 in the GET pathway, transient inactivation of Ssa1 in yeast cells severely disrupted the insertion of GET substrates into the ER membrane, comparable to observations with deletion of GET genes [38].

These findings show that even a compositionally simple integral membrane protein, such as the TA, is sequentially relayed through a multi-component Hsp70-cochaperone cascade before they are loaded on the dedicated protein targeting factor for delivery to the membrane (Figure 4A). Although the complexity of this substrate loading mechanism appears counterintuitive, many intriguing observations during the substrate relays in the GET pathway suggest energetic and organizational principles that may generalize to other Hsp70- and chaperone-mediated protein targeting pathways. These observations and their implications are discussed below.

### 3.1. Client Conformational Quality

Measurements of the stabilities of the different chaperone-TA complexes showed that TAs engage in increasingly stable interactions with chaperones, cochaperones, and targeting factors as they progress along the GET pathway [38,75,79]. This raises the question: if the downstream chaperones bind TAs more tightly, why is participation of Hsp70 necessary? Multiple observations suggest that stepwise substrate loading via Ssa1 significantly enhances the targeting competence of TA substrates. While direct loading of TAs on Sgt2 is inefficient and resulted in largely aggregated, inactive complexes, Sgt2•TA complexes generated via transfer from Ssa1 are not only soluble, but also kinetically competent in undergoing all the subsequent steps in the GET pathway [38]. Although the client interaction of Ssa1 and Sgt2 remain to be understood at molecular detail, kinetic determinants are likely responsible for these observations. The aggregation of single-pass membrane proteins in aqueous environments tends to be rapid and, without external energy input, irreversible [38]. Although Sgt2 and Get3 bind TAs with high affinity, their substrate binding kinetics at physiological concentrations are likely too slow to compete with TA aggregation. In contrast, the rapid client interaction kinetics and high abundance of cytosolic Hsp70 enable it to effectively compete with these off-pathway processes, allowing nascent TAs to be captured and maintained in a soluble, functionally competent conformation.

A recent work showed that the JDP-driven ATPase cycle on Ssa1 further allows it to enhance the targeting competence of TAs [76]. Two major JDPs in yeast, Ydj1 and Sis1, play redundant roles in supporting the insertion of GET substrates into the ER. In vitro, both JDPs enhanced the crosslink of model TA substrates to Sgt2 after the Ssa1-to-Sgt2 TA transfer. The J-domain of both JDPs, which is responsible for activating ATP hydrolysis on Ssa1, was necessary and sufficient for this enhancement. These observations echo recent studies that show that the conformation of client proteins can be actively remodeled by the Hsp70 ATPase cycle, such that they are released in conformations that are more folding-competent [18,80]. Collectively, these studies suggest that not only are the SBDs of Ssa1 and Sgt2 brought into proximity in the TA transfer complex, but the ATPase cycle of Ssa1 further optimizes the conformation of the TA substrate or the Ssa1–Sgt2 complex and thus leads to more optimal Sgt2-TA interactions.

### 3.2. Privileged Substrate Relays

Client handovers between chaperones is ubiquitous and integral to the folding of numerous proteins. For example, substrates resistant to folding by Hsp70 are captured by chaperonins to complete their folding [21,22,23], and outer membrane proteins in bacteria associate with and dissociate from multiple periplasmic chaperones before insertion into the membrane [81]. Studies in the GET pathway provided convincing evidence that TAs are channeled across the chaperone cascade in a strongly facilitated, highly protected mechanism. The Sgt2-to-Get3 (or SGTA-to-TRC40) TA transfer is kinetically facile, with a half-time of 10–20 s [72,75], significantly faster than spontaneous TA dissociation from Sgt2 [38]. In addition, this transfer is protected from off-pathway chaperones, such as calmodulin (CaM) [72,79]. Importantly, such privileged client transfer provides an effective mechanism to not only protect MPs from re-exposure to the cytosol, but also ensure that they are retained within a dedicated biogenesis pathway enroute to the target membrane.

The molecular mechanisms that enable privileged TA handover in the GET pathway is only partly understood. On the one hand, many individual protein domains and interactions in the transfer complex have been extensively studied. In addition to the TPR that connects Sgt2 to the SBD of Hsp70, the N-terminus of Sgt2 mediates its homodimerization and forms the interaction platform for the ubiquitin-like (UBL) domain of Get5 (Figure 4B) [82,83,84], connecting Sgt2 to downstream GET components. As the dedicated TA targeting factor, Get3 is an obligate homodimer in which the ATPase domains bridge the dimer interface and are structurally and functionally coupled to a helical domain (Figure 4B). ATP binding induces ‘closing’ motions that bring together the helical domains, which allows conserved hydrophobic residues in the helical domains to form a contiguous hydrophobic groove that provides the docking site for the TA-TMD (Figure 4B, structural panel d) [85,86,87,88,89]. The scaffolding complex Get4/5 not only brings Sgt2 and Get3 into close proximity, but also bridges the Get3 dimer interface to stabilize Get3 in the closed conformation and inhibits its ATPase activity, thus optimizing the conformation and nucleotide state of Get3 for capture of the TA substrate from Sgt2 (Figure 4B, structural panel d) [90,91,92,93]. A recent study further shows that a conserved structural element (helix α8) lining the substrate-binding groove of Get3, which was not crystallographically resolved, is required for the rapid and privileged TA transfer from Sgt2 to Get3 [79], suggesting a model in which α8 initiates interaction with the TA and guides it into the substrate binding groove of Get3.

On the other hand, how TAs are recognized by Ssa1 and Sgt2 and how they are transferred between these two chaperones remain unclear. The Hsp70 SBD recognizes peptide substrates of 5–7 amino acids, whereas a TA-TMD is ~20 amino acids in length. How Ssa1 effectively protects the TA-TMD is unclear but could be related to the dynamics of the Hsp70-substrate interaction as suggested from recent NMR studies [9,94,95]. The C-terminal SBD of Sgt2 (and SGTA) is rich in glutamine and methionine and shares homology with the SBD of Sti1/HOP. A computational structural model suggests that the Sgt2 SBD folds into a helical hand with a hydrophobic groove that can cradle one face of the TA-TMD (Figure 4B, structural panel c) [73]. This structural model and pull-down experiments also suggest that each SBD in Sgt2 accommodates ~11 amino acids [73], raising questions as to how the TMD is protected by Sgt2. The fact that Sgt2 is an obligate dimer raises the possibility that both of its SBDs engage the TMD, a hypothesis that remains to be tested. Finally, due to the multiple flexible elements in Hsp70, Sgt2, and Get4/5, the relative orientation of the individual domains is unclear, and thus, the architecture of both TA transfer complexes is largely unknown. Answers to these questions are likely key to understand the mechanisms by which TA substrates are channeled in a highly protected manner through the elaborate chaperone cascade in the GET pathway.

### 3.3. Driving Force and Organizational Principles

The complete reconstitution of the GET pathway allowed quantitative biochemical measurements of the individual molecular steps, and the results indicate that this pathway is thermodynamically driven. TA capture by Ssa1 is coupled to JDP-activated ATP hydrolysis in Ssa1, enabling effective TA trapping by this chaperone (Figure 4A, step 3). Both the Ssa1-to-Sgt2 and Sgt2-to-Get3 TA transfers are energetically downhill, with the transfer equilibrium ~100-fold and ~20-fold in favor of the downstream chaperone, respectively (Figure 4A, steps 4–5) [38,75]. Finally, TA binding on Get3 activates the ATPase activity of this targeting factor (Figure 4A, step 6) [92]. ATP hydrolysis induces the detachment of TA-loaded Get3 from Get4/5 and favors its interaction with the Get1/2 receptors instead, driving the movement of the targeting complex from the cytosol to the ER membrane [96,97]. Thus, the GET pathway is accomplished by a sequential series of energetically downhill molecular events.

Another intriguing observation is the highly modular ‘design’ of the GET pathway, in which each chaperone/cochaperone fulfills a distinct and important function. Ssa1, together with the JDPs, captures newly synthesized TAs and ensures their conformational quality, as discussed above (Figure 4A, steps 2–3). The next chaperone in the pathway, Sgt2, appears to serve as an early selection filter in the pathway. The organelle specificity of TA localization is encoded, in part, by a combination of physicochemical properties in the TMD [48,75]. On average, the TMD of mitochondrial TAs are shorter, less hydrophobic, and lower in helical content compared to TAs that are destined to the endomembrane system [75,98,99]. Pulldown experiments in cell lysate showed that Sgt2 preferentially binds TMDs with higher hydrophobicity and helical content, features that distinguish mitochondrial versus ER-destined Tas [75]. A more recent sequence analysis suggests that Sgt2 recognizes one hydrophobic face on the TMD helix [100]. Thus, suboptimal TA substrates for the GET pathway could be rejected by Sgt2 before they are committed to ER targeting and be re-routed to alternative targeting pathways (Figure 4A, steps 4, 7, and 8). As the final chaperone in this cascade and the dedicated targeting factor, Get3 escorts TAs to the receptor/insertase sites at the ER membrane (Figure 4A, step 6). Collectively, the sequential passage of TAs through these distinct functional modules enables the efficient, selective, and unidirectional targeting of nascent TAs to the ER. It will be intriguing to test whether analogous modular designs are used in other post-translational MP targeting pathways and represent a general organization principle.

## 4. Role of Hsp70 in Protein Targeting to Mitochondria

Approximately 900 proteins in yeast and 1100 proteins in human cells are imported into mitochondria [101,102]. Unlike ER-destined proteins that carry a defined targeting signal (Figure 2), mitochondrial proteins reside in at least four different compartments [the outer and inner mitochondrial membranes (OMM and IMM, respectively), the intermembrane space (IMS), and matrix] and carry distinct targeting signals. Roughly 60% of mitochondrial proteins, most of which reside in the mitochondrial matrix, contain an N-terminal mitochondrial targeting sequence (MTS) characterized by an amphiphilic helix comprised of interspersed hydrophobic and basic residues [103,104]. Other mitochondrial proteins use internal MTSs, TMDs, hydrophobic β-hairpins, or cysteine-rich motifs as their targeting signal [103,104]. The majority of mitochondrial proteins are imported via the translocase of the outer mitochondrial membrane (TOM) complex, in which the β-barrel of Tom40 forms the protein-translocation channel (Figure 5A). Recent cryo-electron microscopy (cryoEM) structures [105,106,107,108,109] showed that the core yeast TOM complex consists of two Tom40 pores lined by three small proteins (Tom5, 6, 7) and displays the central preprotein receptor Tom22 (Figure 5A). More dynamically associated with the core TOM complex are two additional preprotein receptors, Tom20 and Tom70, from which mitochondrial preproteins could then access the Tom40 import channel (Figure 5A).

While the composition, function, and mechanism of protein sorting and translocation machineries in mitochondria have been extensively studied [104,110], much less is known about the cytosolic events that deliver preproteins to mitochondria. Compared to the predominantly cotranslational nature of protein targeting to the ER, the majority of mitochondrial proteins are targeted post-translationally. Perhaps, due to the diversity of targeting signals on mitochondrial precursor proteins, no dedicated mitochondrial targeting factor analogous to SRP has been found. Nevertheless, many studies support the extensive involvement of cytosolic chaperones, including Hsp70 and Hsp90, in mitochondrial protein targeting. As described earlier, this notion was initially suggested by early pulse-chase studies of mitochondrial protein import in yeast cells using rapid inactivation of temperature-sensitive Ssa1 or Ydj1 [53,57] and was corroborated in more recent work showing the dependence of mitochondrial protein import on additional JDPs in vivo [111,112,113]. Below, we review biochemical and structural data that suggest a direct involvement of Hsp70 in delivering mitochondrial precursor proteins to receptor sites at the OMM and the cochaperone- and substrate-preference of these delivery pathways.

First, newly synthesized mitochondrial precursor proteins are strongly associated with Hsp70, Hsp90, and their cochaperones. Mitochondrial preproteins translated in yeast lysate or rabbit reticulocyte lysate (RRL) co-purified with Hsp70, Hsp90, three JDPs (Ydj1, Sis1 and Djp1 in yeast, DNAJA1, 2 and 4 in RRL), and Sti1/HOP that bridges between Hsp70 and Hsp90 [114]. While Hsp70 is the most prominent co-purified chaperone in yeast lysate, HSP90 plays a more significant role in the mammalian system [114]. This association was observed with mitochondrial precursor proteins destined to different locales in mitochondria, including multipass MPs and carrier proteins on the IMM [114,115], cysteine-rich proteins in the IMS [114], as well as TAs, signal-anchored MPs (SAs) with a single TMD at the N-terminus, and β-barrel proteins that reside on the OMM [111,113]. On gel filtration chromatography, the chaperone-preprotein assemblies in RRL migrated as a characteristic ~600 kDa complex that contains HSC70, HSP90 and HOP [114,115], suggesting that preproteins are maintained in a soluble complex with multiple chaperones. Notably, a cyclic β-hairpin peptide that mimics the targeting signal on β-barrel OMM proteins was sufficient for interaction with Hsp70 and Ydj1 in vitro, and these interactions were verified in vivo in a photocrosslinking study [113]. Mutations in the J-domain of the JDPs, which interfere with their ability to activate ATP hydrolysis on Hsp70, inhibited preprotein binding to Hsp70, and these defects correlated with reduced import efficiency [115] (Figure 5A, steps 1–2). Analogously, an excess of the C-terminal domain of the NEF BAG-1, which catalyzes substrate release from HSC70, strongly inhibited the import of multiple classes of mitochondrial proteins in RRL [111,113,114]. These observations strongly suggest that the cochaperone-regulated ATPase cycle allows Hsp70 to capture mitochondrial proteins at the earliest stage of their biogenesis and likely maintain the precursor proteins in a soluble, translocation-competent conformation.

Secondly, there is strong evidence that Hsp70 and Hsp90 deliver mitochondrial preproteins to the Tom70 receptor, which dynamically associates with the TOM complex. The cytosolic domain of yeast Tom70 (and a close homologue Tom71 in yeast) contains 11 TPR motifs, which form a superhelix-like structure and can be divided into N- and C-terminal subdomains (Figure 5B). The N-terminal subdomain, comprised of TPR1-TPR3 (Figure 5B,C, *tan*), is homologous to the TPR domain of Sti1/HOP and binds Hsp70 and Hsp90 via their C-terminal EEVD motif (Figure 5A, step 3 and Figure 5B) [114,116,117,118]. The C-terminal fragment of Hsp70 or Hsp90, which competes with the chaperones for interaction with Tom70, inhibited the import of IMM proteins into mitochondria in both yeast lysate and in RRL [114]. This observation was recently extended to TAs, SAs, and β-barrel proteins on the OMM in yeast [111,113,119]. Further, a systems-level study in yeast cells significantly expanded the repertoire of Tom70/71-dependent preproteins and showed that they are enriched in hydrophobic TMDs, internal MTSs, and other aggregation-prone regions [119]. Surprisingly, this study showed that the defects associated with Tom70/71 deletion can be complemented by tethering an unrelated protein on the OMM that contain a chaperone-binding TPR motif [119], supporting the importance of the chaperone-Tom70 interaction in the biogenesis of aggregation-prone mitochondrial proteins.

The C-terminal subdomain of Tom70 (TPR4-TPR11) harbors an extensive hydrophobic pocket that recognizes preproteins [114,116,117,118]. Although MTS sequences could bind in this groove [120], multiple studies show that precursor proteins with multiple internal MTSs preferentially interact with Tom70 and are particularly dependent on this receptor for import [121,122,123]. Client interaction of Tom70 was most recently visualized with the SARS-CoV2 protein Orf9b, which is implicated in immune evasion by targeting TOM70 [124,125,126]. CryoEM and crystallographic structures of the TOM70-Orf9b complex revealed a combination of hydrophobic and salt-bridge interactions that drive their association (Figure 5C) [127]. Notably, this interaction induced the β-sheet rich conformation of Orf9b in solution [128] into an extended α-helix, suggesting a conformational preference that may be relevant to the specificity of preprotein recognition by Tom70 [127]. Importantly, a recent biochemical reconstitution provided evidence that a β-hairpin peptide pre-bound to Ssa1 could be transferred to the Tom70 cytosolic domain [111], suggesting a potential path for substrate handover that remains to be tested (Figure 5A, step 4). Whether and how the preproteins bound to Hsp70 or Tom70 are further transferred to the TOM complex for import across the OMM remain outstanding questions.

Notably, Tom70 specifically associates with Djp1 [129], an abundant yet poorly studied JDP in yeast that is localized in the cytosol as well as on mitochondria and the ER surface. Both Djp1 and Tom70 play important roles in the biogenesis of Mim1 and Mim2, which belong to a class of proteins that are anchored on the OMM by an internal TMD. A yeast genetic screen identified only two genes, Djp1 and Tom70, that are strongly implicated in Mim1 biogenesis [130]. The deletion of Djp1, but none of the other JDPs in yeast, reduced the steady-state levels of Mim1 and led to its mislocalization at the ER [130]. In addition, in vitro translated Mim1 co-purified with Djp1 and Hsc70, and with Tom70 when incubated with mitochondria, indicating physical interactions [130]. However, the precise mechanism by which Hsp70 participates in Tom70- and Djp1-dependent Mim1 import and the role of the specific association of Djp1 with Tom70 remains unclear. It is also not known whether these roles of Djp1 and Tom70 are related, in part, to a recently described ER surface retrieval (ER-SURF) pathway in which Djp1 retrieves mitochondrial proteins initially localized to the ER surface and re-routes them to mitochondria [131].

In contrast to Tom70, Tom20 preferentially recognizes MTS-containing precursor proteins, although the two receptors display partially overlapping substrate specificity [111,114,132]. The NMR structure of the receptor domain of yeast Tom20 showed that its surface presents a hydrophobic groove that binds the hydrophobic face of the MTS helix [77,133,134]. The substrate preference of Tom20 is shared by Tom22 [132], the central receptor in the core TOM translocase that presents MTS binding sites in both of its cytosol and IMS-facing domains. It is unclear whether preproteins bound to Tom20 were transferred to Tom22 for import through the TOM channel. Of note, the JDP Xdj1 was found to strongly and specifically associate with Tom22 in vivo and in vitro [129]. Deletion of Xdj1 in yeast led to reduced levels of the TOM complex, and mitochondria from xdj1Δ cells displayed reduced import efficiency, although it is unclear whether these observations reflect a direct or indirect effect of Xdj1 loss [129]. Surprisingly, Xdj1 specifically co-purified with in vitro translated mitochondrial proteins containing hydrophobic recognition segments, but not with hydrophilic mitochondrial precursor proteins [129]. Partially purified substrate proteins bound to Xdj1 could be transferred to the cytosolic domain of Tom22 and was competent for import into isolated mitochondria [129]. The J-domain of Xdj1 was not required for its binding to Tom22 but was essential for stimulating protein import into mitochondria, implicating Hsp70 in Xdj1-dependent protein import. It is not clear whether Xdj1 and Tom22 act downstream of Hsp70/Tom70 or provide an Tom70-independent pathway for the import of hydrophobic mitochondrial proteins, nor what the precise molecular events are in this pathway.

## 5. Additional Hsp70 Involvements in Protein Targeting

Hsp70 was also implicated in the targeting of proteins to peroxisomes but with far less evidence. The peroxisome targeting signal type 1 (PTS1) receptor, Pex5, belongs to the TPR motif family and mediates the translocation of PTS1-containing proteins to peroxisomes. An early study using PTS1-containing protein translated in RRL suggested that substrate binding by Pex5 is dependent on ATP and enhanced by Hsp70, which binds to the TPR domain in Pex5 [135]. Moreover, protein import into peroxisomes is inhibited by Hsp70 depletion [135]. However, these results were disputed in a later study, which showed that Pex5 can bind PTS1-containing substrates with high affinity and independently of Hsp70 and nucleotides [136]. In addition, the import of proteins containing the PTS2 peroxisome-targeting signal was also found to be dependent on Hsp70, Hsp40, and ATP [137]. Whether the observed stimulatory effects of Hsp70 can be attributed solely to its ability to mediate the proper folding or solubilization of peroxisomal proteins or to a more direct role in targeting remains an outstanding question.

## 6. Conclusions

In addition to cytosolic protein substrates, Hsp70 is remarkably effective in capturing newly synthesized MPs and organellar proteins and in maintaining their targeting competence. The interaction of Hsp70 with TPR-containing targeting factors and receptor complexes further enables this chaperone to direct the targeted delivery of these proteins to diverse translocation machineries on organellar membranes. Detailed mechanistic studies in the GET pathway highlight how Hsp70 can initiate a series of substrate relays in a sophisticated chaperone cascade that effectively and selectively funnels hydrophobic TAs to dedicated insertases at the ER membrane. Hsp70 and its cochaperones also play key roles in the biogenesis of nascent mitochondrial proteins. While the role of Tom70 as the major Hsp70 receptor is well established, much of the molecular details of how Hsp70-bound mitochondrial proteins are delivered to the TOM translocase for import remain to be understood. The observations with Djp1 and Xdj1 further suggest the presence of multiple pathways via which Hsp70 can participate in mitochondrial protein import, the mechanism of which remain outstanding questions. Finally, Hsp70 is also implicated in the targeting of secretory and GPI-anchored proteins to the ER [138] and in the import of peroxisomal proteins, but its role in these processes remains to be defined.

## 7. Additional Outstanding Questions

*What provides the specificity of protein targeting when the same Hsp70s engage with proteins destined to different organelles?* This question is especially intriguing given the observation that the major housekeeping Hsp70 and JDPs in yeast, Ssa1 and Ydj1/Sis1, respectively, are involved in the targeting of TAs to both the ER and mitochondria. It is possible that specificity is conferred by the ability of the downstream cochaperone/receptor to receive client proteins from Hsp70, but this hypothesis remains to be tested.*What are the NEFs associated with each of the distinct Hsp70-mediated targeting pathways?* As described in Section 1, NEFs are an essential component of the Hsp70 ATPase cycle during protein folding and remodelling. Although the roles of Hsp70 and JDPs in protein targeting has been extensively documented, little is known about whether NEFs participate in Hsp70-dependent protein targeting pathways and if so, what their precise roles are. The observations that a model substrate protein could be rapidly transferred from Ssa1 to a downstream cochaperone (Sgt2) or receptor (Tom70) in purified systems raises the possibility that NEFs may not be required to drive substrate release from Hsp70 during protein targeting. However, NEFs may still play critical roles in returning Hsp70 to the ATP-state for additional rounds of protein targeting. Both of these possibilities remain to be tested.*Are Hsp70s involved in Hsp40-dependent protein targeting pathways?* As described in Section 4, in multiple cases the involvement of Hsp40 in mitochondrial protein targeting has been described, but the involvement of Hsp70 in these processes was unclear. In many of these cases, testing the effect of mutations in the signature HPQ motif of the J-domain, which is essential for inducing ATPase activation in Hsp70, would provide key answers to this question and may uncover additional Hsp70-JDP pairs that participate in protein targeting.*What happens to protein targeting under different environmental conditions?* Besides the housekeeping Hsp70s, many Hsp70 isoforms and paralogs are induced by proteostatic stress, such as heat shock. In addition, the aggregation propensity of MPs and organellar proteins in the cytosol is likely exacerbated under stress conditions. This raises the possibility that Hsp70-dependent protein targeting pathways could be regulated under different environmental conditions. The fate of nascent post-translationally targeted proteins and how the Hsp70 network is rewired to handle them during proteostatic stress remain outstanding questions.

## Figures and Tables

**Figure 1 ijms-24-01170-f001:**
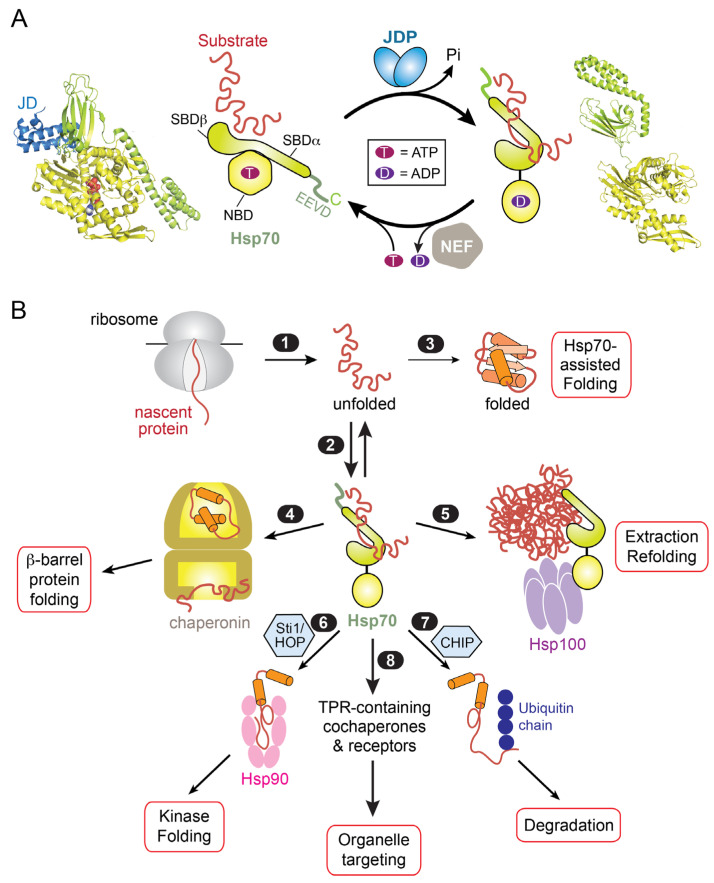
Overview of the Hsp70 functional cycle and its role in protein biogenesis. (**A**) The ATPase cycle of Hsp70 regulates the capture and release of substrate proteins and is driven by its cochaperones J-domain proteins (JDPs) and nucleotide exchange factors (NEFs), which catalyze the hydrolysis of ATP and the release of ADP, respectively. The various domains in Hsp70 are indicated. The left panel shows the crystal structure of the J-domain (JD) of the bacterial JDP, DnaJ (*blue*), bound to the bacterial Hsp70 DnaK in the ATP-state (PDB# 5nro). The right panel shows the crystal structure of DnaK in the apo-state (PDB# 2kho). The SBD and NBD of DnaK are in *lime* and *lemon*, respectively. The bound ATP is in *spacefill*. (**B**) Hsp70 acts as a central hub in the cellular chaperone network to coordinate diverse protein biogenesis pathways, as described in the text.

**Figure 2 ijms-24-01170-f002:**
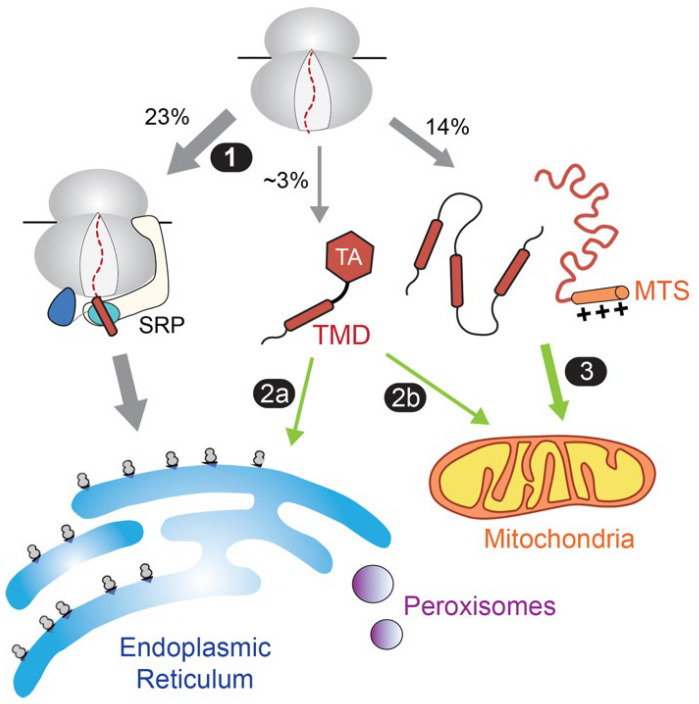
Summary of Hsp70 involvement in protein targeting in eukaryotic cells and their relationship to cotranslational protein targeting to the ER. Membrane proteins destined to the endomembrane system, including the ER, Golgi, secretory vesicles, and plasma membrane, are primarily targeted to the ER membrane by the cotranslational SRP pathway (path 1). TAs are targeted to the ER or mitochondria post-translationally (paths 2a and 2b). The majority of mitochondrial proteins, including α-helical and β-barrel membrane proteins as well as soluble proteins destined to the mitochondrial matrix, are targeted via post-translational mechanisms (path 3). The targeting of proteins to peroxisomes also occurs post-translationally, but the involvement of Hsp70 is unclear. The values represent estimated percentages in the yeast proteome [37]. Green arrows indicate Hsp70 involvement in the targeting paths. MTS, mitochondrial targeting sequence.

**Figure 3 ijms-24-01170-f003:**
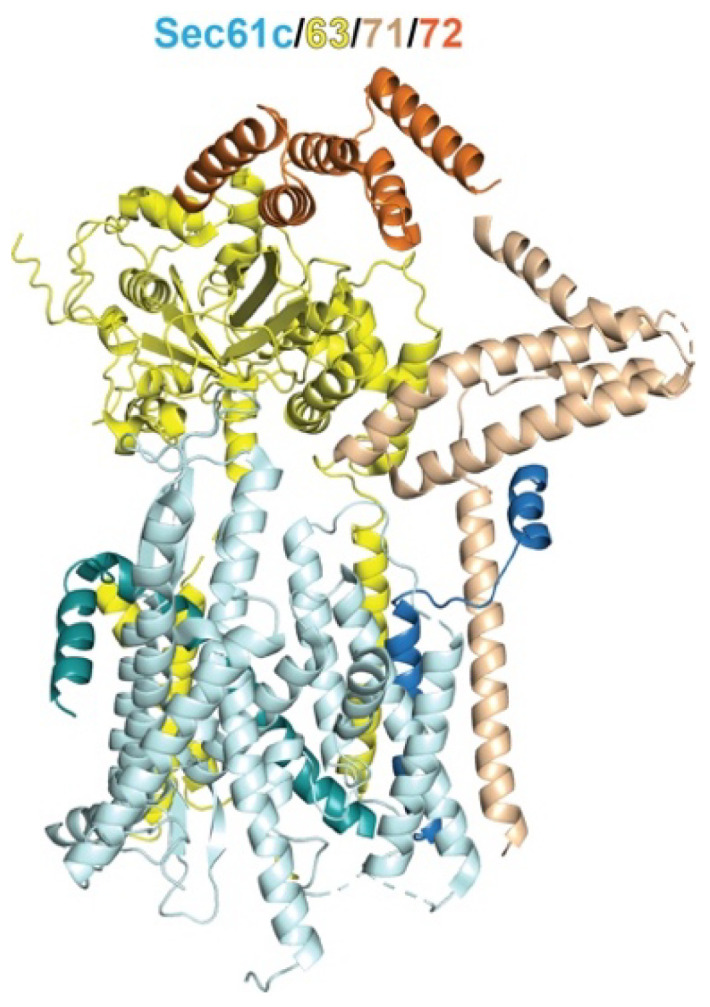
The cryoEM structure of the yeast Sec61/63/71/72 complex (PDB# 6nd1; [64]) for post-translational protein translocation across the ER membrane. The TPR domain of Sec72 is highlighted in *orange.* Sec61c denotes the yeast Sec61 complex, which forms the core translocation channel and consists of the protein subunits Sec61 (*aqua*), Sbh1 (*cambria*), and Sss1 (*teal*) (which correspond to mammalian SEC61α, β, and γ, respectively).

**Figure 5 ijms-24-01170-f005:**
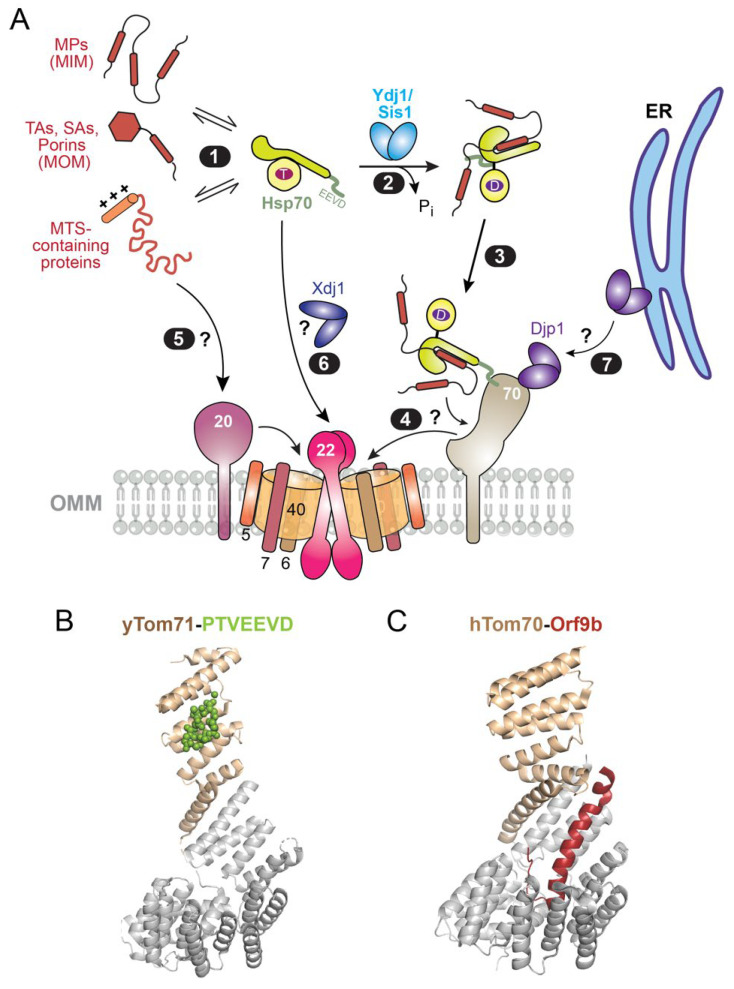
Role of Hsp70 in protein targeting to mitochondria. (**A**) Potential cytosolic events during the targeting of mitochondrial proteins. Diverse mitochondrial proteins with distinct targeting signals and biophysical properties are captured by Hsp70, likely with assistance of JDPs (steps 1–2). Substrate-loaded Hsp70 can associate with the Tom70 receptor on the mitochondria surface via interaction of its EEVD motif with the N-terminal TPR domain of Tom70 (step 3). Hsp70-bound substrates might be transferred to the preprotein-binding groove on Tom70 or directly to the core TOM complex (step 4). MTS-containing precursor proteins are preferentially captured by the Tom20 receptor (path 5). The JDP Xdj1 specifically associates with Tom22 in the core TOM complex and assists in mitochondrial protein import in an Hsp70-dependent manner, the mechanism of which is unclear (path 6). Additionally, the JDP Djp1 specifically binds Tom70 and is important for the biogenesis of Mim1; it is unclear whether this role of Djp1 is related to its ability to retrieve mitochondrial proteins localized to the ER membrane (path 7). (**B**) The crystal structure of Tom71 cytosolic domain bound to the Hsp70 C-terminal peptide (*lime*; PDB 3fp4). The N- and C-terminal subdomains of Tom71 are in *tan* and *grey*, respectively. (**C**) The crystal structure of human Tom70 bound to the Orf9b peptide (*brick*; PDB 7dhg). Domain coloring is the same as in (**B**).

## Data Availability

Not applicable.

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
