# Peer review of "Role of Hsp70 in Post-Translational Protein Targeting: Tail-Anchored Membrane Proteins and Beyond"

_ijms, 2023, doi:10.3390/ijms24021170_

Round 1

Reviewer 1 Report

The article by Shu-ou Shan reviews the literature on targeting to both the ER and mitochondrial membranes, from an Hsp70-mediated perspective. The work is timely given the number of new papers recently published on the topic and the important outstanding questions in the field. The figures were well-done and will be helpful to the community. Overall, the review largely focussed on Tail Anchor (TA) protein targeting, and from this point of view, the review was comprehensive, thoughtful and generally well-written. However, the content of the review is not appropriately reflected by the title: in many cases there were long discussions about non-Hsp70-mediated targeting details, and also sections where much more is known about the Hsp40s/JDPs. On the other hand, some aspects of Hsp70 mediated targeting were not reviewed in appropriate depth. A change in title (to be more TA protein centric) or addition/re-organisation of some sections would help make the article more cohesive and structured. In addition, a few sentences could be written more accurately for clarity. However, following the below changes I strongly support publication of this manuscript that will be helpful and informative to the many people working on protein targeting to organelles.

Specific comments and requests are found below divided into three sections:

1) typos/small textual changes

·      In general, the first time abbreviates are used they should be spelled out in full. For example: HSP, HSC, TMD, and all other abbreviations including amino acids (eg: EEVD).

·      Throughout the manuscript, please go through each of the sections and explain which system/organism is being used (specifically for protein names and percentages or numbers of proteins). Currently, there is a lot of interchange between yeast and mammalian systems).

·      In general, there are many points in the review where it is unclear whether a TA protein or a TA domain is being referred to. Please refer to the protein as a ‘TA protein’ and only the TMD as a TA.  

·      Line 12: the comma should be a hyphen or colon

·      Line 18: missing a full stop at the end of the sentence

·      Line 23: “protein homeostasis” could also be referred to as “proteostasis”

·      Line 27: “associated” should be “associate”

·      Lines 28-31: Hsp70s also recognise client oligomers that are correctly folded and assembled but simply need to be dissociated such as Hsf1 trimers and assembled clathrin. They are also required for pulling nascent proteins into organellar lumens (eg: BIP in the ER and the Hsp70 of the mitochondrial matrix). Please modify these sentences to reflect these functions.

·      Lines 36-37: “exposed in client proteins as they deviate from the native structure” – this is not always true. Please update.

·      Line 39: packs against the NBD

·      Line 41: client proteins

·      Line 47: Hsp70s

·      Line 56: ‘structure of the J-domain (JDP)…of the bacterial JDP’ – only the J-domain protein needs abbreviating to JDP.

·      Line 64 – ‘resistant’ might be more understandable that ‘refractory’

·      Line 66 – mitochondria also have a GroEL/ES (Hsp60). ‘Hsp70’ should be ‘Hsp70s’.

·      Line 79 – ‘protein trafficking’ should be ‘biogenesis’ as trafficking would suggest packing into vesicles

·      Line 86-88 – it’s not very clear why this is relevant since luminal proteins are soluble and this sentence is sandwiched between sentences on membrane proteins. If is it included, please also make clear which system/organism you got these numbers (>1000) from and also include the lumen of other organelles such as lysosomes and Golgi.

·      Line 94 – it is unclear how rapid and/or reversible aggregation is, especially without the addition of a reference.

·      Line 99 – as above, there is no reference to support the claim that the majority of ER-targeted proteins are targeted co-translationally and at least in yeast this is also not the case. Please add reference supporting this or modify accordingly.

·      Line 102 – make it clear what recognises TMDs and/or SPs as it is not only SRP. When talking about SRP you should reference the original Walter and Blobel paper/s.

·      Line 104 – a lot of proteins are not shuttled to Sec61 according to recent data from the Hegde lab. Please also acknowledge the EMC here.

·      Line 126 – although the GET pathway is the best-studied for TA protein ER targeting and translocation, the SND pathway in yeast and humans can also mediate this process and the EMC is a well-established TA protein insertase. These should therefore be noted.

·      Line 130 – ‘abundance’ would be more widely understood than ‘preponderance’

·      Lines 136-141 – data from Hsp70 KOs is also difficult to interpret because of the presence of homologous back-up pathways.

·      Lines 164-167 – Get1-3 were found in the referenced paper (#45), but Get4 and Get5 as well as Sgt2 were found later: https://pubmed.ncbi.nlm.nih.gov/19325107/

·      Line 179 – ‘associates’ should be ‘associate’

·      Lines 206-207 - please mention this: https://pubmed.ncbi.nlm.nih.gov/33542241/

·      Lines 215-217 – it seems possible that this could occur via the membrane-bound fraction of Ydj1.

·      Line 250 – ‘tends to be rapid’ – is there a reference for this?

·      Line 258 – ‘play redundant roles’ – Sis1 and Ydj1 may have overlapping roles in this capacity, but they are not redundant proteins.

·      Lines 263-264 – not sure what needs to be folding-competent….the rest of the protein beside the TA?

·      Line 266 – ‘lead’ should be ‘leads’

·      Line 270 – ‘recalcitrant’ is not a commonly used word and a more accessible one should be considered.

·      Lines 270-273 – make clear that this is now in bacteria.

·      Line 299 – ‘show’ should be ‘shows’

·      Line 330 – recent work discussing the post-hydrolysis Get3 state could be referenced: https://www.nature.com/articles/s41594-022-00798-4.

·      Line 331 – what is the reference for favouring Get1/2 interaction?

·      Line 361 – ‘residing’ should be ‘reside’

·      Line 363 – ‘comprised of’?

·      Line 364 – MTSs

·      Lines 367-370: not clear how relevant this is to the review and probably should be removed to stay on focus with the review topic.

·      Line 371 – Tom71 in yeast should also be mentioned.

·      Line 399 – this is not believed to be true: https://pubmed.ncbi.nlm.nih.gov/25378625/

·      Line 406 – it would also be good to discuss what determines specificity/directionality.

·      Line 413 – ‘locale’ should be ‘locales’

·      Line 422 – ‘studies’ should be ‘study’

·      Lines 475-477 – it would be good to spell out that this is called the ER-SURF pathway and note that the identity of the Hsp70 is not known.

·      Lines 545-546 – other J-domain proteins have been implicated in this: https://pubmed.ncbi.nlm.nih.gov/23452858/

·      Line 547 – please add a reference for the Hsp70-mediated import into peroxisomes.

·      Line 782-783 – this reference did not appear in the text.

·      Some recent work which was not included in the review: https://journals.plos.org/plosbiology/article?id=10.1371/journal.pbio.3001839 https://journals.plos.org/plosgenetics/article?id=10.1371/journal.pgen.1010442

2) larger textual/conceptual changes

·      Given the earlier work on the role of Hap70 chaperones with the auxiliary translocon, starting with this topic instead of leaving it to the very end of the manuscript as a “side note” may be better representative of the history of the field and could work better (and figure 4D could then be a stand-alone for this section as it is right now inserted into a mitochondrial-centric figure).

·      Make sure to correctly discuss J protein actions as "specificity determinants". All JDPs with a bound substrate have been shown to also bind HSP70s in the ATP-bound state (in cells the JDPs are 10 time less abundant than HSP70s) and thus recruit ATP-Hsp70s onto the protein complexes to which the JDPs are specifically bound. So they clearly have a much more central and important role in client determination than conveyed in this review. The model should look more like this: the concomitant binding of ATP-HSP70 to JDP-substrate protein complexes, causes ATP hydrolysis in HSP70. This causes the closure of the HSP70 lid and the local unfolding, by clamping, on the bound proteins, and also unfolding by entropic pulling, and the concomitant dissociation of the J-domain from the ADP-Hsp70 for which is has a very low affinity.

·      It would be good to discuss the presence of other Hsp70s that have been suggested to have a role (or at least a backup role) in targeting such as Ssa2,3,4?

·      The addition of an ‘outstanding questions/future directions’ section would be very beneficial to this article. Future questions could include: What provides directionality to targeting when the same Hsp70s are working with ER, Mitochondria and peroxisomal proteins? What are the NEFs associated with each of the distinct HSP-mediated pathways? The HSPs are stress induced – so what happens to targeting under stress/different conditions? Also interesting is that in many cases, as highlighted throughout the review, the Hsp40 has been found, but not the Hsp70. Some comment/insight into this would be useful.

3) figure-associated changes

·      In general:

o   Unifying the colour-schemes throughout the figures would help the reader. In addition unifying the way a protein or TMD are drawn is important – sometimes in the same figure (For example Figure 1B panels 4 and 6 each have a protein drawn differently).

o   Please make the names of colours either in italics or not (currently there is a mix of both).

o   It would be better not to call the numbered routes in the figures, ‘steps’, because, with the exception of Figure 3A, these are not consecutive events. Instead, call them ‘paths’ as was done for Figure 2.

·      Figure 1A – better labelling would make this easier to follow. Specifically: label the JDP in the structure; label substrate with full word, not just an S; indicate C’ of the Hsp70. The ATP molecule is so small that it should just be coloured in one colour (red).

·      Figure 1B - #5 should also say ‘degradation’, #7 should be illustrated since it is the only one without an associated cartoon.

o   Drawing the ribosome with a nascent chain emphasises the fact that ribosomes can engage at the exit tunnel. This doesn’t necessarily have to be included in this figure, but it should at least be acknowledged in the text (eg: https://www.nature.com/articles/s41467-021-25930-8)

·      Figure 2 – the title of the figure (first sentence in the legend) does not reflect what the figure shows since there are many more targeting pathways which are not shown here. In this figure, it would be good to point out how the HSP proteins fit in. It is also not clear where the numbers come from and what they mean (23% and 14%? It all adds up to 40% but what is this 40%?). Please add the appropriate references to the legend.

o   In the legend, the SRP pathway is not the only way (see comment on line 102 above and adjust accordingly). Peroxisomes should be mentioned in the context of post-translational targeting.

·      Figure 3 – where is the ADP in 5? Here, the labelling of the TA is confusing – please point to the TMD of the TA protein. Not sure if ‘step 1’ is necessary.

o   In general, is it correct to call this a Hsp70-cochaperone cascade (title of figure and within the main text) since no Hsp40 is mentioned here?

·      Figure 4 – (A) Some MIM proteins also have MTSs; some multipass proteins are also on the MOM; IMS proteins usually do not have MTSs, but rather CXC motifs (having an MTS is really an exception not the rule). (D) This figure panel cannot be grouped with a figure called ‘targeting to mitochondria’. When mentioning Sec61, Sbh1 and Sss1 make clear this is in yeast and say they correspond to SEC61α, β, γ.

Author Response

Point-by-point response to reviewer comments are listed below.

Reviewer 1:

  1. A change in title (to be more TA protein centric) or addition/re-organisation of some sections would help make the article more cohesive and structured.

The title has been modified to “Role of Hsp70 in post-translational protein targeting: tail-anchored membrane proteins and beyond”.

1) typos/small textual changes

  • In general, the first time abbreviates are used they should be spelled out in full. For example: HSP, HSC, TMD, and all other abbreviations including amino acids (eg: EEVD).

We have gone through the manuscript and defined all the abbreviations. For example, HSP and HSC are defined. “EEVD motif” was changed to “motif with the amino acid sequence EEVD”.

  • Throughout the manuscript, please go through each of the sections and explain which system/organism is being used (specifically for protein names and percentages or numbers of proteins). Currently, there is a lot of interchange between yeast and mammalian systems).

We have gone through all the sections to define whether the observations were made in yeast or mammalian systems. We also used a convention where the sentence case is used to refer to yeast proteins, and the uppercase is used to refer to mammalian proteins. This is explained (line 324-325) and will hopefully clarify some of the confusion, especially in the section on the GET pathway.  

  • In general, there are many points in the review where it is unclear whether a TA protein or a TA domain is being referred to. Please refer to the protein as a ‘TA protein’ and only the TMD as a TA.  

‘TA’ is defined in the manuscript as ‘tail-anchored membrane protein’ (line 124). Throughout the text, we refer to proteins as ‘TA’ or ‘TA substrate’. We refer to the TMD of the TA explicitly as ‘TA-TMD’ or ‘the TMD of TA’.

  • Specific editing suggestions:

‘ü’ below denotes that the suggestion has been incorporated. Where the topic warrants additional nuanced discussions, additional responses are described.

ü      Line 12: the comma should be a hyphen or colon

ü      Line 18: missing a full stop at the end of the sentence

ü      Line 23: “protein homeostasis” could also be referred to as “proteostasis”

ü      Line 27: “associated” should be “associate”

ü      Lines 28-31: Hsp70s also recognise client oligomers that are correctly folded and assembled but simply need to be dissociated such as Hsf1 trimers and assembled clathrin. They are also required for pulling nascent proteins into organellar lumens (eg: BIP in the ER and the Hsp70 of the mitochondrial matrix). Please modify these sentences to reflect these functions.

ü     Lines 36-37: “exposed in client proteins as they deviate from the native structure” – this is not always true. Please update.

ü      Line 39: packs against the NBD

ü      Line 41: client proteins

ü      Line 47: Hsp70s

ü      Line 56: ‘structure of the J-domain (JDP)…of the bacterial JDP’ – only the J-domain protein needs abbreviating to JDP.

ü      Line 64 – ‘resistant’ might be more understandable that ‘refractory’

ü      Line 66 – mitochondria also have a GroEL/ES (Hsp60). ‘Hsp70’ should be ‘Hsp70s’.

  • Line 79 – ‘protein trafficking’ should be ‘biogenesis’ as trafficking would suggest packing into vesicles

Changed to ‘protein targeting’.

  • Line 86-88 – it’s not very clear why this is relevant since luminal proteins are soluble and this sentence is sandwiched between sentences on membrane proteins. If is it included, please also make clear which system/organism you got these numbers (>1000) from and also include the lumen of other organelles such as lysosomes and Golgi.

Organellar proteins in the cytosol are, by definition, in a non-native environment where they may not reach their correct 3D structure. This is most obvious for proteins that undergo oxidative folding in the ER lumen, but the principle extends to other organellar proteins. Even if some the organellar proteins are soluble in the cytosol, prolonged residence in the cytosol will define them as mislocalized proteins that are subject to quality control.

For simplicity, the phrase is revised to ‘misfolding, aggregation, and degradation’.

The lumen of other organelles are added.

  • Line 94 – it is unclear how rapid and/or reversible aggregation is, especially without the addition of a reference.

Example references added.

  • Line 99 – as above, there is no reference to support the claim that the majority of ER-targeted proteins are targeted co-translationally and at least in yeast this is also not the case. Please add reference supporting this or modify accordingly.

References added.

  • Line 102 – make it clear what recognises TMDs and/or SPs as it is not only SRP. When talking about SRP you should reference the original Walter and Blobel paper/s.

I am confused by this comment. It is well established that the N-terminal TMDs and hydrophobic signal sequences emerging from the ribosome are recognized by SRP before they are targeted to the ER membrane.  If the reviewer are thinking about another factor that recognize ER targeting signals, please name it specifically.

Walter and Blobel paper referenced.

  • Line 104 – a lot of proteins are not shuttled to Sec61 according to recent data from the Hegde lab. Please also acknowledge the EMC here.

Sentence modified to include ‘other insertases’.

We note that even in the recent work from Hegde lab (Smalinskaite et al, Nature 2022) that studies the interaction of cotranslationally translocating nascent chain with a number of other membrane protein chaperone/insertases using cryoEM and crosslinking, the best structurally resolved component of the translocating complex (other than the ribosome) is Sec61p. The objective interpretation of these data is that Sec61p is the primary ribosome docking site that anchors the translocating ribosome-nascent chain complex at the ER, and other translocases or MP chaperones are dynamically assembled to the vicinity of Sec61p to interact with the nascent chain upon need.

ü     Line 126 – although the GET pathway is the best-studied for TA protein ER targeting and translocation, the SND pathway in yeast and humans can also mediate this process and the EMC is a well-established TA protein insertase. These should therefore be noted.

ü      Line 130 – ‘abundance’ would be more widely understood than ‘preponderance’

ü     Lines 136-141 – data from Hsp70 KOs is also difficult to interpret because of the presence of homologous back-up pathways.

ü     Lines 164-167 – Get1-3 were found in the referenced paper (#45), but Get4 and Get5 as well as Sgt2 were found later: https://pubmed.ncbi.nlm.nih.gov/19325107/

ü      Line 179 – ‘associates’ should be ‘associate’

ü      Lines 206-207 - please mention this: https://pubmed.ncbi.nlm.nih.gov/33542241/

  • Lines 215-217 – it seems possible that this could occur via the membrane-bound fraction of Ydj1.

The biochemical data is very strong that Ssa1 and the JDPs Ydj1/Sis1 act upstream of Sgt2, Get3, Get4/5, and Get1/2. As described in section 3, the transfer of TA across these components is strongly unidirectional (i.e., not reversible). Moreover, the action of Ydj1 in the GET pathway is dependent on Ssa1, as deleting the client-binding CTDs in Ydj1 does not abolish its role in facilitating TA loading on Sgt2, whereas mutating the JD of Ydj1 that activates Ssa1 does. Since Ssa1, Sgt2 and Get4/5 are primarily cytosolic, TA does not reach the ER until it is loaded on Get3.  It is therefore unlikely that ER-bound Ydj1 plays a role in this process. 

  • Line 250 – ‘tends to be rapid’ – is there a reference for this?

TMD aggregation in aqueous solution is rooted in the hydrophobic effect. Studies of protein folding have placed hydrophobic collapse on the 10-100 ns timescale (doi:10.1042/BCJ20160107). Nobody has studied TMD aggregation kinetics, because aggregation is not compatible with most rapid mixing instruments. However, references showing that TA aggregation is complete within manual mixing time is added.

  • Line 258 – ‘play redundant roles’ – Sis1 and Ydj1 may have overlapping roles in this capacity, but they are not redundant proteins.

The observation was that depletion of either JDP had no effect on TA insertion into the ER, but depletion of both severely reduced TA insertion. The statement (“play redundant roles in supporting the insertion of GET substrates into the ER’) refers specifically to TA targeting, for which ‘redundant’ is accurate.

  • Lines 263-264 – not sure what needs to be folding-competent….the rest of the protein beside the TA?

Folding competence refers to preventing the entire TA from inappropriate inter- or intramolecular interactions. While there is some evidence that Sgt2 primarily engages with the TA-TMD, the interaction site of Hsp70 on the TA is unclear. Although the TA TMD is a strong candidate, some TAs have unstructured N-terminal regions where Hsp70 interaction cannot be excluded.  In the absence of data we prefer to refer to the entire TA substrate.

ü      Line 266 – ‘lead’ should be ‘leads’

ü      Line 270 – ‘recalcitrant’ is not a commonly used word and a more accessible one should be considered.

ü       Lines 270-273 – make clear that this is now in bacteria.

ü      Line 299 – ‘show’ should be ‘shows’

ü      Line 330 – recent work discussing the post-hydrolysis Get3 state could be referenced: https://www.nature.com/articles/s41594-022-00798-4.

ü      Line 331 – what is the reference for favouring Get1/2 interaction?

References added.

ü     Line 361 – ‘residing’ should be ‘reside’

ü      Line 363 – ‘comprised of’?

ü      Line 364 – MTSs

  • Lines 367-370: not clear how relevant this is to the review and probably should be removed to stay on focus with the review topic.

Lines 367-370 serve to introduce the three major mitochondrial preprotein receptors, Tom20, Tom70, and Tom22 that are described extensively in Section 4. They also place these receptors in the context of the TOM import machinery and explain the membrane-associated components depicted in Figure 4A.  Edited to remove the last sentence and improve flow.

  • Line 371 – Tom71 in yeast should also be mentioned.

Tom 71 is described later on Line 660 in the paragraph dedicated to describing the Tom70 receptor.

  • Line 399 – this is not believed to be true: https://pubmed.ncbi.nlm.nih.gov/25378625/

Add reference.

If one look at the gene-level enrichment score in Figure 1C and 1D of this paper, it is clear that a subset of mitochondrial proteins are targeted while being translated, but the majority is not. Post-translational mitochondrial protein targeting is still the primary targeting route for mitochondrial proteins. The statement (“the majority of mitochondrial proteins are believed to be targeted post-translationally”) is accurate.

  • Line 406 – it would also be good to discuss what determines specificity/directionality.

There are no data on the directionality of mitochondrial protein targeting, which requires quantitative studies of the energetics of preprotein interaction with Hsp70 and with the mitochondrial preprotein receptor. The last sentence of the paragraph is extended to include the cochaperone- and substrate preference of targeting routes.

ü      Line 413 – ‘locale’ should be ‘locales’

ü     Line 422 – ‘studies’ should be ‘study’

ü      Lines 475-477 – it would be good to spell out that this is called the ER-SURF pathway and note that the identity of the Hsp70 is not known.

ü      Lines 545-546 – other J-domain proteins have been implicated in this: https://pubmed.ncbi.nlm.nih.gov/23452858/

  • Line 547 – please add a reference for the Hsp70-mediated import into peroxisomes.

References for Hsp70 involvement in peroxisomal protein import are in section 5. Section 6 is the ‘conclusion’ section, which is not appropriate to repeat the citations. Otherwise we will also need to insert almost all the references for GET, mitochondrial targeting, etc.

  • Line 782-783 – this reference did not appear in the text.

Cited in the legend of new Figure 3.

  • Some recent work which was not included in the review: https://journals.plos.org/plosbiology/article?id=10.1371/journal.pbio.3001839https://journals.plos.org/plosgenetics/article?id=10.1371/journal.pgen.1010442

            This link is broken (‘Page not found”).

  • Given the earlier work on the role of Hap70 chaperones with the auxiliary translocon, starting with this topic instead of leaving it to the very end of the manuscript as a “side note” may be better representative of the history of the field and could work better (and figure 4D could then be a stand-alone for this section as it is right now inserted into a mitochondrial-centric figure).

The paragraph is moved to lines 228-245. Figure 4D is moved to be new Figure 3.

  • Make sure to correctly discuss J protein actions as "specificity determinants". All JDPs with a bound substrate have been shown to also bind HSP70s in the ATP-bound state (in cells the JDPs are 10 time less abundant than HSP70s) and thus recruit ATP-Hsp70s onto the protein complexes to which the JDPs are specifically bound. So they clearly have a much more central and important role in client determination than conveyed in this review.

JDPs are extremely diverse and include classes A, B, and C. Classes A and B JDPs have CTDs that serve as general peptide binding sites and class C JDPs mediate specific interaction with protein complexes to recruit Hsp70s for protein complex remodeling. The term ‘specificity determinant’ is only appropriate for class C JDPs. Except for the J-domain that activates ATP hydrolysis on Hsp70s, many of the other functions of JDPs are not universal. For examples directly related to this article, Ydj1 and Sis1 are the major cytosolic JDPs in yeast and are required for the targeting of GET substrates to the ER as well as the targeting of TAs, SAs, and hydrophobic b-barrel proteins to mitochondria. Neither of them can be described as ‘specificity determinants’ in their targeting role. Djp1 and Xdj1 may more specifically recruit Hsp70 to the ER or mitochondrial surface, but this has yet to be experimentally shown. We therefore refrained from making sweeping statements about JDPs’ functions other than the core function of the universally conserved J-domain, as such statements will be confusing or misleading without going into further detail about the different classes of JDPs. We only amended the paragraph to include a statement about the diversity of JDPs and how their pairing allows Hsp70s to remodel diverse client proteins (lines 75-78).

The model should look more like this: the concomitant binding of ATP-HSP70 to JDP-substrate protein complexes, causes ATP hydrolysis in HSP70. This causes the closure of the HSP70 lid and the local unfolding, by clamping, on the bound proteins, and also unfolding by entropic pulling, and the concomitant dissociation of the J-domain from the ADP-Hsp70 for which is has a very low affinity.

Suggested changes are incorporated with minor editing.

  • It would be good to discuss the presence of other Hsp70s that have been suggested to have a role (or at least a backup role) in targeting such as Ssa2,3,4?

A statement about the potential redundant roles of Ssa1-4 is included (lines 225-227).

  • The addition of an ‘outstanding questions/future directions’ section would be very beneficial to this article. Future questions could include: What provides directionality to targeting when the same Hsp70s are working with ER, Mitochondria and peroxisomal proteins? What are the NEFs associated with each of the distinct HSP-mediated pathways? The HSPs are stress induced – so what happens to targeting under stress/different conditions? Also interesting is that in many cases, as highlighted throughout the review, the Hsp40 has been found, but not the Hsp70. Some comment/insight into this would be useful.

New section has been added as suggested.

3) figure-associated changes

  • In general:
  • Unifying the colour-schemes throughout the figures would help the reader. In addition unifying the way a protein or TMD are drawn is important – sometimes in the same figure (For example Figure 1B panels 4 and 6 each have a protein drawn differently).

Consistent color schemes have been used throughout as much as possible. Substrate proteins are always in brick, with TMD helices represented as rounded squares in brick and soluble protein a-helices as orange cylinders.

  • Please make the names of colours either in italics or not (currently there is a mix of both).

Checked and corrected.

  • It would be better not to call the numbered routes in the figures, ‘steps’, because, with the exception of Figure 3A, these are not consecutive events. Instead, call them ‘paths’ as was done for Figure 2.

The current nomenclature is correct in Figure 4 (new Figure 5). Steps 1-4 are sequential steps in a path to deliver hydrophobic proteins to the TOM translocase. Numbered routes 5-7 are referred to as ‘paths’ as they denote additional potential pathways independent of those depicted by 1-4..

  • Figure 1A – better labelling would make this easier to follow. Specifically: label the JDP in the structure; label substratewith full word, not just an S; indicate C’ of the Hsp70. The ATP molecule is so small that it should just be coloured in one colour (red).

Suggested changes are incorporated except for the bound ATP. It is in spacefill for better visibility and is not small compared to most of the text in the figure, and the cpk coloring helps distinguish the nucleoside base (blue/white) vs the g-phosphate (orange/red) part of the molecule.

  • Figure 1B - #5 should also say ‘degradation’, #7 should be illustrated since it is the only one without an associated cartoon.

#5 – revised as suggested.

#7 – kept as is because essentially the words were expanded into Figures 4A and 5A and is the subject of the majority of the review.

  • Drawing the ribosome with a nascent chain emphasises the fact that ribosomes can engage at the exit tunnel. This doesn’t necessarily have to be included in this figure, but it should at least be acknowledged in the text (eg: https://www.nature.com/articles/s41467-021-25930-8).

The point that Hsp70s can interact with nascent proteins during their synthesis is incorporated (line 30).

We want to avoid going into further detail on this topic, because this is a complex topic worthy of an entire review article in itself. Cotranslational Hsp70 engagement is not restricted to Ssb’s but also to DnaK in bacteria and mammalian HSC70 and Hsp70. Among them, there are distinctions as to whether the engagement is mechanistically cotranslational (meaning interaction with ribosome is obligatory) or temporally cotranslational (meaning the interaction occurs during translation). This topic should either be discussed with substantive text space, taking into account all the nuances and current data, or be omitted. It was a conscious writing choice to skip this topic and focus on post-translational protein targeting and Hsp70’s involvement in them, as this review was commissioned to do.  As we understand, this chaperone is one in a series that reviews diverse aspects of Hsp70, and other aspects of Hsp70 functions and interactions is likely covered by other reviews in this series.

  • Figure 2 – the title of the figure (first sentence in the legend) does not reflect what the figure shows since there are many more targeting pathways which are not shown here. In this figure, it would be good to point out how the HSP proteins fit in. It is also not clear where the numbers come from and what they mean (23% and 14%? It all adds up to 40% but what is this 40%?). Please add the appropriate references to the legend.

The point of Figure 2 is not to comprehensively describe all the known protein targeting pathways but to highlight proteins that may be targeted by Hsp70 to different organelles (green arrows) and to contrast the post-translational nature of Hsp70-dependent targeting with the established cotranslational protein targeting pathway via SRP. The title of Figure 2 legend has been revised accordingly.

The numbers represent estimated percentage in the yeast proteome. Citation is provided. 

  • In the legend, the SRP pathway is not the only way (see comment on line 102 above and adjust accordingly). Peroxisomes should be mentioned in the context of post-translational targeting.

Removed ‘secretory protein’ from the legend. SRP is the primary pathway to target MPs to the ER and the remainder of the endomembrane system.

  • Figure 3 – where is the ADP in 5? Here, the labelling of the TA is confusing – please point to the TMD of the TA protein. Not sure if ‘step 1’ is necessary.

ADP legend has been removed. The TMD of TA is labeled. ‘step 1’ is kept because this makes it easier to refer to in the text and in figure legend.

  • In general, is it correct to call this a Hsp70-cochaperone cascade (title of figure and within the main text) since no Hsp40 is mentioned here?

Sgt2 is an Hsp70 cochaperone, analogous to sti1/HOP. Hsp40 is also involved in TA targeting and regulates the ability of Hsp70 to load TA on Sgt2, as described in the text and shown in Figure 4.

  • Figure 4 – (A) Some MIM proteins also have MTSs; some multipass proteins are also on the MOM; IMS proteins usually do not have MTSs, but rather CXC motifs (having an MTS is really an exception not the rule).

This figure summarizes the types of proteins that have been studied as Hsp70 substrates for their targeting.  We have changed IMS/matrix proteins to ‘MTS-containing proteins’, but otherwise the information is correct.

(D) This figure panel cannot be grouped with a figure called ‘targeting to mitochondria’. When mentioning Sec61, Sbh1 and Sss1 make clear this is in yeast and say they correspond to SEC61α, β, γ.

Fig. 4D has been moved to become new Figure 3. Figure legends are edited as suggested.

Reviewer 2 Report

The Hsp70s are doubtless the most promiscuous family of molecular chaperones as they are involved in all different aspects of protein homeostasis (folding, disaggregation and degradation) and also in different “gain of function” processes. There are a large number of reviews covering these topics but the review by Dr. Shu-ou Shan deals with a new and in my opinion interesting field, that of the role of Hsp70 in the post-translational targeting of proteins (mostly membrane proteins but also soluble ones) to different cellular organelles (ER, mitochondria and peroxisomes). Dr. Shan’s review comprehensible describes what is known about these processes in which Hsp70 seems to have a key role, but which is supported by a plethora of other chaperones, cochaperons and specialized receptor proteins. The review has a very interesting structural edge because the information is accompanied by several figures which show in structural terms what is known about the coordinated mechanisms of targeted delivery of nascent proteins to a particular cellular membrane. In my opinion, the review should be published in its current form.

A minor comment: please correct in Figure 1B “b-barrel protein folding”

Author Response

Reviewer 2:

The Hsp70s are doubtless the most promiscuous family of molecular chaperones as they are involved in all different aspects of protein homeostasis (folding, disaggregation and degradation) and also in different “gain of function” processes. There are a large number of reviews covering these topics but the review by Dr. Shu-ou Shan deals with a new and in my opinion interesting field, that of the role of Hsp70 in the post-translational targeting of proteins (mostly membrane proteins but also soluble ones) to different cellular organelles (ER, mitochondria and peroxisomes). Dr. Shan’s review comprehensible describes what is known about these processes in which Hsp70 seems to have a key role, but which is supported by a plethora of other chaperones, cochaperons and specialized receptor proteins. The review has a very interesting structural edge because the information is accompanied by several figures which show in structural terms what is known about the coordinated mechanisms of targeted delivery of nascent proteins to a particular cellular membrane. In my opinion, the review should be published in its current form.

A minor comment: please correct in Figure 1B “b-barrel protein folding”

Thanks. Corrected.